# Genome-Wide Identification of the TIFY Family in Longan and Their Potential Functional Analysis in Anthocyanin Synthesis

**DOI:** 10.3390/biology14040364

**Published:** 2025-04-01

**Authors:** Haowei Qing, Ziang Wu, Xiao Mo, Jinjv Wei, Yuyu Shi, Huiqin Guo, Jiongzhi Xu, Feng Ding, Shuwei Zhang

**Affiliations:** 1Guangxi Key Laboratory of Genetic Improvement of Crops/Guangxi Academy of Agricultural Sciences, Nanning 530007, China; 18777200815@163.com (H.Q.); 18376683951@163.com (X.M.); qt15797862018@163.com (J.W.); sy1045731981@163.com (Y.S.); huigin@163.com (H.G.); 2College of Agriculture, Guangxi University, Nanning 530007, China; wza67690@163.com (Z.W.); xujiongzhi-2@163.com (J.X.); 3Institute of Horticulture, Guangxi Academy of Agricultural Sciences, Nanning 530007, China

**Keywords:** *Dimocarpus longan*, TIFY proteins, sequence analysis, anthocyanin synthesis

## Abstract

The color change of the pericarp of *Dimocarpus longan* is not obvious, lack of visual appeal greatly limits the value of the product. Therefore, it is of great significance to study the regulation of anthocyanin synthesis in longan. In recent years, the TIFY gene family has been reported in many species, proving that it plays an important role in the accumulation of anthocyanins in fruits. In this study, 19 members of the Longan TIFY transcription factor family were screened and identified through the genome database of longan. It was demonstrated that *DlTIFY7* played a negative regulatory role in the anthocyanin synthesis pathway by transgenic tobacco. *DlTIFY7* may be a new candidate transcription factor for negative regulation of anthocyanin synthesis of longan. The aim of this study was to provide a theoretical basis for the synthesis of anthocyanins of longan and to provide a new target gene for breeding and improvement of longan varieties.

## 1. Introduction

TIFY transcription factors are a plant-specific class that contains the TIFY domain with a core motif of TIF[F/Y]XG. Based on conserved sequence domains, they are classified into four subfamilies: TIFY, PEAPOD (PPD), JASMONATE ZIM-Domain (JAZ), and ZIM-LIKE (ZML). The TIFY subfamily only contains the TIFY domain, whereas the other three subfamilies have specific domains alongside the TIFY domain. The PPD subfamily contains the TIFY domain, PPD domain, and JAS domain, lacking the conserved PY residue. The JAZ subfamily comprises a TIFY and a JAS domain. Within the ZML subfamily, proteins have a TIFY domain, accompanied by a CCT domain and GATA zinc finger domain. The first TIFY family genes were identified in *Arabidopsis thaliana* known as zinc finger protein ZIM expressed in inflorescence meristematic tissues [1]. To date, 18 *TIFY* genes have been systematically analyzed in *Arabidopsis*. High-throughput technology has enabled genome-wide investigation of the TIFY family genes. So far, TIFY transcription factors have been found in several plants, such as bananas, tomatoes, apples, grapes, watermelons, pears, and loquat, providing an important foundation for functional studies of them.

TIFY protein was widely involved in regulating plant growth and development, stress, and hormone signaling [2]. The first characterized *AtTIFY1* gene was involved in the regulation of inflorescence, flowering, and petioles and hypocotyl extension [3], while *AtTIFY4a* and *AtTIFY4b* were involved in the process of leaf development [4]. *OsTIFY11b* and *OsTIFY3* were identified as regulators of grain size and spikelet development in rice [5,6]. The TIFY proteins also play a role in abiotic stress and phytohormone responses. *AtJAZA7* was reported to produce drought tolerance by regulating photosynthesis, plant hormones and defense metabolites. Overexpression of the *OsTIFY11a* gene in rice could significantly improve tolerance to salt and drought stress [7]. As reported, the pear TIFY proteins are potentially positively or negatively correlated with anthocyanin accumulation. Several *TIFY* genes in Chinese sand pear was induced after MeJA and light treatment in the red fruit [8], while *PpZML1* and *PpZML3* showed a negative correlation with fruit epicarp anthocyanin contents [9]. *MdJAZ2* negatively regulated anthocyanin accumulation by repressing the expression of the anthocyanin regulatory and structural genes in apple [10].

Longan (*Dimocarpus longan Lour.*) is an important economic fruit of the Sapindaceae family and is widely cultivated in tropical and subtropical regions of Southeast Asia. Longan, with its high nutritional, economic, and medicinal value, has a long history of cultivation in China [11]. During the development and ripening of longan peel, the color of the peel changes from green to yellowish brown, which is mainly the result of chlorophyll degradation and anthocyanin accumulation. Compared with litchi, longan peel color change is not obvious; lack of visual appeal greatly limits the value of the product. Therefore, it is of great significance to study the regulation of anthocyanin synthesis of longan. Recently, studies on the TIFY gene family have been reported in many species, which proves that it plays an important role in the accumulation of anthocyanins in fruit [10]. However, the TIFYs in longan plants remain poorly researched. The completion of longan genome sequencing enables the genome-wide identification of the TIFY gene family. Herein, in the present study, members of the TIFY transcription factor family in longan were screened and identified through the longan genome datasets. The conserved motifs, gene structure, evolutionary relationships, and cis-acting elements were investigated. Further, we analyzed the protein interaction network of DlTIFY7. Consequently, subcellular localization and functional analysis of DlTIFY7 was performed. Our research represents an initial investigation of the functional characterization of longan *TIFY* genes, thereby enriching the theory of the regulation of anthocyanin biosynthesis. This research provides a foundation for further exploration of the functions of TIFY factors in longan.

## 2. Materials and Methods

### 2.1. Identification and Sequence Analysis of DlTIFY Genes

The genome and gene annotation files of ‘Shixia’ longan was supplied by SapBase (XIALAB from the college of horticulture at South China Agricultural University; genome size 483 Mb; Gene Number 37142; accessed on 24 July 2024). The hidden Markov model (HMM) profiles of the TIFY domain (PF062000) (https://www.ebi.ac.uk/interpro/; accessed on 24 July 2024) were used as queries to search the DlTIFY proteins by TBtools (Version V2.154) [12]. The protein sequences were submitted to Pfam (https://www.ebi.ac.uk/interpro/; accessed on 24 July 2024), SMART (http://smart.embl-heidelberg.de; accessed on 24 July 2024), and cd-search (https://www.ncbi.nlm.nih.gov/Structure/cdd/wrpsb.Cgi; accessed on 24 July 2024) for identification of TIFY-conserved functional domains. Relative molecular weights and isoelectric points were predicted in the ExPASy ProtParam tool (https://www.expasy.org/; accessed on 24 July 2024). At the same time, subcellular location prediction of TIFY proteins were generated with Plant-mPLoc (http://www.csbio.sjtu.edu.cn/bioinf/plant-multi/; accessed on 24 July 2024).

The gene structure provides important information, including disaggregated and evolutionary relationships among gene families. The *DlTIFY* genomic sequences and coding DNA Sequence (CDS) were extracted from the genome database to determine the exon–intron structure information of *TIFY* genes. Default parameters were used for the Multiple Em for Motif Elicitation (MEME) (http://meme-suite.org/; accessed on 25 July 2024) online tool for the prediction of conserved protein motifs and a maximum number of 10 motifs. Default settings were adopted and the results imported into TBtools for visualization.

### 2.2. Sequence Alignments and Phylogenetic Analyses

Multiple alignments of TIFY protein sequences from longan, grape (*Vitis vinifera*), orange, rice, and *Arabidopsis* were performed by ClustalW with default parameters. The TIFY protein sequences were downloaded from the ensembl website (http://asia.ensembl.org/index.Html; accessed on 25 July 2024), The phylogenetic tree was constructed with the MEGA11.0 software using the maximum likelihood method with default parameters.

### 2.3. Chromosomal Location and Gene Duplication

The physical location data of the *DlTIFY* genes were generated from the longan genome. The position of these *TIFY* genes was exhibited using TBtools (Version V2.154) software. Aanlysis of genome collinearity of longan was performed using MCScanX software (Version V2.154). Gene duplication events of *DlTIFY* genes in longan genome were analyzed and visualized using TBtools software (Version V2.154), with E-value cut-off of <1^−10^ (Version V2.154).

### 2.4. Analysis of Cis-Acting Elements of DlTIFY Genes

According to the genome sequences of ‘Shixia’ longan, the upstream 2000 bp of the 5′ sequence was used to analyse the *cis-acting* elements by the online software PlantCARE (https://bioinformatics.psb.ugent.be/webtools/plantcare/html/; accessed on 25 July 2024). The results were visualized by the TBtools program (Version V2.154).

### 2.5. Expression Patterns of DlTIFY Genes

The RNA-seq data of *DlTIFY* family members in different tissues and organs were downloaded from the SapBase database [13]. TBtools software (Version V2.154) was used to visualize the expression levels of *DlTIFYs*.

### 2.6. Subcellular Location and Protein Interaction Network Analysis of DlTIFY7

The pAN580 vector was linearized by XbaIand BamHI double enzyme digestion. The cDAN of longan leaves is used as the template; coding the sequence of *DlTIFY7* without the termination codon was cloned into the pAN580 vector for fusion with the green fluorescent protein GFP under control of the 35S promoter. The constructed plasmid was transformed into GV3101 strains to infiltrate epidermal cells *Nicotiana benthamiana*. After extracting the protoplast, the fluorescence of GFP was observed by laser scanning confocal microscopy at the wavelength of excitation light at 488 nm and reflected light at 510 nm (Nikon C2-ER, Nikon Microsystems, Shanghai, China). The primers are listed in Appendix A. The specific transformation steps of protoplasts are referred to the experimental method of Sang-Dong et al. [14]. The DlTIFY7 protein interaction network was predicted based on the corresponding homologous protein in *Arabidopsis* in the STRING web server (https://string-db.org/; accessed on 12 September 2024).

### 2.7. Overexpression of DlTIFY7 in Tobacco

To over-express *DlTIFY7*, the full-length coding sequence of *DlTIFY7* was inserted into pSAK277 vector under control of CaMV 35S promoter. This construct was transformed into K326 tobacco plants using the *Agrobacterium-mediated* transformation method. The positive transgenic plants were confirmed by PCR amplification. The plants were grown at 28 °C under 16 h/8 h light/dark cycle conditions.

## 3. Results

### 3.1. Genome-Wide Identification of DlTIFY Genes and Chromosomal Location

To identify TIFY family genes in *D. longan*, HMM profile and BLASTp search were performed. Totally, 19 *TIFY* genes were identified and named *DlTIFY1*–*DlTIFY19* according to their positions on the chromosomal (Table 1). The length of CDS ranged from 411 bp (*DlTIFY7*) to 2160 bp (*DlTIFY19*). The protein lengths ranged from 136 to 719 amino acids, with pI ranging from 4.69 to 9.60. Subcellular localization predictions by the Plant-mPLOC tool showed that, except for DlTIFY19, which was located in the cytoplasm and nucleus, the remaining 18 DlTIFYs were all in the nucleus. The 19 *DlTIFY* genes were irregularly mapped to nine chromosomes (Chr) of longan.

### 3.2. Conserved Domain and Phylogenetic Analysis of the DlTIFY Proteins

The conserved domain location was showed in Figure 1. All the 19 DlTIFY proteins contained the TIFY domain, and were further categorized into three subfamilies: TIFY, JAZ, and ZML. DlTIFY3 contains only one TIFY domain, suggesting it belongs to the TIFY subfamily. Six proteins belonged to the ZML subfamily and were designated as DlTIFY8 to DlTIFY12 and DlTIFY18. The remaining 12 proteins have a TIFY and a Jas domain belonging to the JAZ subfamily. To understand the evolutionary relationship of the DlTIFY with other species, an ML phylogenetic tree was constructed (Figure 2). The phylogenetic analysis revealed that the 61 TIFYs, from *Arabidopsis*, rice, grape, orange, and longan, were classified into six groups. The DlTIFY3 was clustered together with AtTIFY8 and VvTIFY8 in group 2. All six DlTIFY proteins of the ZML subfamily were clustered in group 1. The 12 DlTIFY proteins of the JAZ subfamily were distributed across four other groups. Most DlTIFY members displayed a closer relationship to the *Citrus sinensis*.

### 3.3. Conserved Motif and Gene Structure Analysis

The 19 *DlTIFY* genes were irregularly mapped to nine chromosomes of longan (Appendix A). In total, 10 motifs were identified, named motif 1 to motif 10 (Figure 3A). Motif 1 (TIFY motif) was distributed in all the 19 DlTIFY members. Genes belonging to the same phylogenetic group have similar motifs structures. There were significant differences in gene structure among members of the longan *TIFY* gene family (Figure 3B), with the number of introns ranging from 1 to 18, among which *DlTIFY7* contained 1 intron and *DlTIFY19* contained 18 introns.

### 3.4. Analysis of Cis-Acting Elements of DlTIFY Genes

*Cis-acting* elements in gene promoters are crucial regions for gene expression and revealing the differences in transcript expression and biological function. The upstream 2000 bp sequence of 19 *DlTIFY* genes were analyzed by the PlantCARE software (Figure 4). All members of the TIFY gene family in longan had the light responsiveness element. Many elements were related to hormones and stress in the promoter of *DlTIFY* genes. Excluding *DlTIFY3* and *DlTIFY8*, all other *DlTIFY* genes had abscisic acid responsive element.

There were also many hormone-related cis-acting elements, e.g., MeJA-, ABA-, gibberellin (GA)-, salicylic acid (SA)-, and auxin (IAA)-responsive. In particular, the flavonoid biosynthetic genes regulation element was distributed in the promoters of *DlTIFY7*, *DlTIFY10*, and *DlTIFY11*.

### 3.5. Chromosome Distribution and Gene Duplication Analysis of DlTIFY Genes

Nineteen *DlTIFY* genes were unevenly mapped to nine chromosomes of longan (Appendix A). The largest number of four genes was found in Chromosome (chr) 1 and chr 9. Chromosomes chr 10 and chr 11 contained three *DlTIFY* genes each. Chromosomes chr 3, chr 4, chr 7, chr 8, and chr 15 contained only one *DlTIFY* each. To further determine the amplification and evolution of *DlTIFY* genes, the duplication events were analyzed using MCScanX. The results showed that two pairs of genes (*DlTIFY9/DlTIFY10, DlTIFY11/DlTIFY12*) underwent tandem repeat events on Chr 9. Three segmentally duplicated pairs were found by collinearity analysis of the *DlTIFY* genes. Pair DlTIFY1/5 was located on chromosomes 1 and 3, respectively. Pair *DlTIFY4/6* was located on chromosomes 1 and 4, respectively. Pair *DlTIFY16/17* was located on chromosome 11 (Figure 5A). These three gene pairs had a Ka/Ks ratio of 0.243734153, 0.251109463390483, and 0.34793564233152, respectively. The Ka/Ks ratio is less than 1, suggesting that these three *TIFY* gene pairs were driven by purifying selection pressures during the duplication process.

To investigate the composition and evolutionary relationships among the TIFY proteins, a comparative syntenic analysis was performed on the Sapindaceae fruit tree, including longan, litchi (*Litchi chinensis*), and rambutan (*Nephelium lappaceum*). According to the comparative syntenic map, a total of 15 *DlTIFY* genes exhibited collinearity with 14 litchi genes, and 16 *DlTIFY* genes showed syntenic relationships with 15 rambutan genes, respectively (Fighre 5B). We also found that several *DlTIFY* genes showed collinearity with multiple litchi and rambutan genes. In total, 19 and 21 pairs of collinear genes were identified with litchi and rambutan, respectively. *DlTIFY10*, *DlTIFY12*, and *DlTIFY14* had no co-located genes with any of litchi or rambutan, suggesting that these genes may have formed after the plant differentiation of the three fruit tree in Sapindaceae. Ten *DlTIFY* genes, *DlTIFY2*, *DlTIFY3*, *DlTIFY7*, *DlTIFY8*, *DlTIFY11*, *DlTIFY13*, *DlTIFY15*, *DlTIFY16*, *DlTIFY17*, and *DlTIFY19*, were collinear to the unique litchi and rambutan gene each, indicating a good collinearity in longan, litchi, and rambutan. The results indicate that these TIFYs are functionally conserved.

### 3.6. GO and KEGG Enrichment Analysis of DlTIFY Genes

To further understand the biological functions, gene function annotation of the 19 *DlTIFY* genes was performed (Figure 6). The results showed that 17 of the 19 *DlTIFY* genes were annotated and assigned to molecular function and biology process categories. The enrichment of *DlTIFY* genes for transcription coregulator activity, transcription corepressor activity, transcription regulator activity, protein–macromolecule adaptor activity, and molecular adaptor activity were higher among the molecular functions, indicating that *DlTIFY* TF family genes play a vital role in transcription regulation. Among the biological processes, *DlTIFY* members were involved in jasmonic acid signaling and response. In addition, KEGG pathway analysis revealed that 8 of the 19 *DlTIFY* genes were enriched in plant hormone signal transduction, namely, *DlTIFY1*, *DlTIFY2*, *DlTIFY5*, *DlTIFY13*, *DlTIFY14*, *DlTIFY15*, *DlTIFY16*, and *DlTIFY17*.

### 3.7. The Protein–Protein Interaction Network Analysis of the TIFY Family in Longan

Using protein–protein interactions to connect unknown functional proteins into protein interaction networks will help understand the biological functions of proteins. In this study, *Arabidopsis* was used as a background to predict the potential interacting proteins associated with the protein function of DlTIFYs. The orthologous STRING proteins with the highest bit score were identified using all DlTIFY proteins. According to statistical analysis, a total of 10 DlTIFYs found their positions in the interaction network (Figure 7A). Among the 10 detected proteins, DlTIFY3 had the most interaction partners, followed by DlTIFY7. An interaction network was constructed based on an Arabidopsis association model. The results showed that the DlTIFY7 protein has relational interactions with MYC, MYB, and WRKY transcription factors, such as DlMYC2, DlMYB305, DlWRKY51, with the protein ID of Dil.12g012620, Dil.12g011220, and Dil.06g018040 in the SapBase database, respectively (Figure 7B). DlMYC2, DlMYB305, and DlWRKY51 shared homology with AtMYC2 (At1g32640), AtMYB24 (At5g40350), and AtWRKY51 (At5g64810). These interaction proteins were mainly involved in jasmonic acid (JA) signaling pathways.

### 3.8. Expression Pattern of DlTIFY Genes in Different Tissues

To explore the possible functions of *DlTIFY* genes, the expression pattern of the 19 *DlTIFYs* in 9 various tissues of longan were characterized based on the transcriptomes data in SapBases. Genes with low expression levels in all samples were filtered out. The results showed the expression variation of *DlTIFY* genes in different tissues (Figure 8). The expression patterns of most genes were the same in both varieties. Eight genes (*DlTIFY1*, *DlTIFY2*, *DlTIFY5*, *DlTIFY13*, *DlTIFY14*, *DlTIFY15*, *DlTIFY18*, and *DlTIFY19*) were highly expressed in leaves. *DlTIFY3* was predominantly expressed in the pulp. The expression of *DlTIFY8* was higher in the seed tissues. In addition, *DlTIFY7* showed high expression levels in the stem, followed by the pericarp. *DlTIFY4*, *DlTIFY9,* and *DlTIFY11* were most abundantly transcribed in the flower buds of ‘Shixia’; however, they showed highest expression in the roots, flowers, and leaves of ‘Sijimi’, respectively. *DlTIFY6* showed highest expression in the pulp of ‘Shixia’, while in the flower buds of ‘Sijimi’. These results indicated that *DlTIFYs* exhibited tissue expression specificity.

### 3.9. DlTIFY7 Is Localized to the Nucleus and Overexpression of DlTIFY7 Inhibits Anthocyanin Biosynthesis in Transgenic Tobacco

Protein–protein interaction network analysis showed that the DlTIFY7 protein interacts with MYC, MYB, and WRKY proteins, and MYB is a key enzyme in the anthocyanin synthesis pathway. The expression patterns of DlTIFY7 in different tissues showed that DLTIFY7 was highly expressed in the pericarp of longan, indicating that DLTIFY7 was probably involved in the synthesis of secondary metabolites in the pericarp. To further study the function of the DlTIFY7 protein, we performed a subcellular localization experiment and functional verification experiment of transgenic tobacco for this gene.

To determine the subcellular localization of DlTIFY7 proteins, the coding sequences without termination codons of the two genes were fused to a vector carrying GFP. The control GFP protein produced fluorescence signals throughout the cell, whereas the Pro35S:DlTIFY7-GFP fusion protein was detected exclusively in the nucleus. The results indicate that DlTIFY7 is a nuclear-localized protein (Figure 9A), which was consistent with the subcellular localization prediction.

Previous studies have implicated TIFYs in the regulation of plant development, response to abiotic stress, and plant hormones. To further identify the biological function of *DlTIFY7*, we constructed an overexpression vector harboring *DlTIFY7* driven by the 35S promoter and used it to transform wild-type K326 tobacco by *A. tumefaciens*-mediated transformation. The DlTIFY7-OE transgenic lines exhibited no significant variation compared to the control plants throughout the vegetative growth period. However, it was discovered that the color of the flowers were lighter than that of the control group during flowering (Figure 9B). We found that a slightly green color was visible in the corolla tube of transgenic tobacco. Moreover, there was a green ring around the edge of the corolla. It is speculated that this gene negatively regulates anthocyanin synthesis and promotes chlorophyll synthesis.

## 4. Discussion

In this study, a total of 19 *DlTIFY* genes were identified in the longan genome using BLAST and HMMER search. Consistent with observations in other plant species, the JAZ subfamily was the largest among the TIFY subfamilies in longan. DlTIFY3, which encodes a protein only containing a TIFY structural domain, belonged to the TIFY subfamily, 12 proteins with CCT_2 structural domain addition to TIFY domain belonged to the JAZ subfamily, and the remaining six protein belonged to the ZML subfamily; no PPD subfamily gene was found in the longan genome. This supports the idea that not all subfamilies exist in each species [15]. It is hypothesized that the PPD subfamily was found exclusively in dicotyledons and absent in monocotyledonous plants [7,15]. However, one *PaPPD* gene was identified in *Phalaenopsis* recently, which indicates that PPD also exists in moncots. In fact, unlike other dicot plants, which have all four subfamilies, such as peach, apple, loquat, the PPD subfamily was absent in longan. The *PPD* gene may be lost during the divergence of eudicots. PPD proteins have a role in cell cycle and cell proliferation regulation in *Arabidopsis* [4]; other genes in longan are most likely compensating for the molecular activities performed by *PPD* genes. The number of *DlTIFY* family genes in longan was similar to that in other fruit tree species, such as grape (19), apple (16), peach (*Prunus persica*, 16), and kiwifruit (*Actinidia chinensis*, 21) [9,16,17,18]. It is less than loquat, strawberry, blueberry. The number of the TIFY family may be related to both genome size and gene duplication or loss during the process of evolution.

In this study, subcellular localization prediction results showed that, except for DlTIFY19, which is located in the nucleus and cytoplasm, other members of the DlTIFY gene family are located in the nucleus, which is consistent with the mechanism that transcription factors need to enter the nucleus to play a regulatory role. The localization of transcription factors in the nucleus and cytoplasm may be caused by the fact that transcription factors are synthesized in the cytoplasm and have nuclear localization signals (NLS) and nuclear export signals (NES). When the transcription factor is not activated, it may bind to the output protein through NES and stay in the cytoplasm. When the cell receives a specific signal, the transcription factor is phosphorylated and undergoes other modifications, resulting in the exposure of NLS or the masking of NES, so that it can enter the nucleus and play a role. *TIFY* family genes showed variability in gene size and structure in many plant species. In the present study, members of the DlTIFY family are diverse in their sequence structure, similar to the *TIFY* genes in rice [7], grapevine [16], and kiwifruit [17]. The amino acid ranged from 136 aa to 719 aa, while the variation range of introns was 1 to 19. Introns perform various functions and influence gene evolution [19]. Those variations implying functional affection of their structure change during evolution.

Gene duplication events play a crucial role in functional divergence among genes and expedite the emergence of novel genes. Segmental duplication and tandem duplication can lead to the expansion of gene families [20,21]. In this study, two tandem duplications were identified in the DlTIFY family, which included the gene pairs of *DlTIFY9/DlTIFY10* and *DlTIFY11/DlTIFY12*. In addition, three duplication events were also identified with the Ka/Ks values less than one. A comparative syntenic analysis was conducted in comparison with litchi and rambutan. The analysis revealed that 10 *DlTIFY* genes are collinear to the unique genes of litchi or rambutan. This finding strongly indicates the conservation of them within the Sapindaceae fruit trees, suggesting potential common genetic characteristics and their function.

Previous studies have shown that the gene expression pattern likely reflects its biological role. In *Arabidopsis thaliana*, *AtTIFY8* is highly expressed in the roots and is involved in regulation of leaf senescence. Meanwhile *AtTIFY6a* was highly expressed in the stems, witch involved in freezing tolerance and JA iduceed leaf senesence [22,23]. In rice, most of the *OsTIFY* genes were highly expressed in leaves, and several *OsTIFY* genes were expressed highly in the sheath, panicles, or stamen. Almost all OsTIFY genes in rice respond to one or more abiotic stresses, including drought, salinity, and low temperatures. The *OsTIFY11a* gene was mainly expressed in the roots and young ears of rice. Overexpression of the *OsTIFY11a* gene can improve the drought and salt tolerance of rice [7]. In this study, the expression pattern of *DlTIFY* genes was characterized. These genes display tissue-specific expression patterns. *DlTIFY1*, *DlTIFY2*, *DlTIFY5*, *DlTIFY13*, *DlTIFY14*, *DlTIFY15*, *DlTIFY18,* and *DlTIFY19* were highly expressed in the leaves of ‘Shixia’ and ‘Sijimi’ longan. This suggests that they may play an important role in leaf development, metabolism, environmental response, or hormone regulation, while *DlTIFY3* was predominantly expressed in the root and pulp. This means that *DlTIFY3* may play a role in abiotic stresses such as salt stress and drought stress. The pulp is an important place for the accumulation of sugars, organic acids, and secondary metabolites. *DlTIFY3* may be involved in the regulation of these metabolic pathways and affect the flavor and nutritional value of fruits. *DlTIFY7* is highly expressed in the pericarp of ‘Shixia’ and ‘Sijimi’ longan. Pericarp is the main site for the synthesis of secondary metabolites such as anthocyanins, flavonoids, and phenolic compounds, and *DlTIFY7* may be involved in the synthesis pathway of secondary metabolites. Expression analysis results suggest that the functions of these genes have diverged over evolutionary time. The protein–protein interaction network further implies that these DlTIFY proteins may exert their functions through the formation of heterodimer. Specifically, DlTIFY7 is located in the nucleus and exhibits the highest expression levels in the stem, followed by pericarp. Co-expression network analysis suggests that DlTIFY7 may function as a transcription factor and form a heterodimer with DlMYC2, DlMYB305, or DlWRKY51 to regulate the transcription of downstream genes in longan. Their homologs in *Arabidopsis*, *AtMYC2*, *AtMYB24,* and *AtWRKY51*, were involved in JA response [24,25,26,27]. JA is a plant hormone that induces the formation and accumulation of secondary metabolites. Promoter cis-elements of the gene promoter are often utilized to predict the potential function. Various elements that deal with stress response, hormone response, and development were identified in the promoter sequence of *TIFY* genes. Previous studies showed that the MYB binding site involved in drought stress and regulation of the flavonoid biosynthetic genes was observed in the promoter of *TIFY* genes [17,28,29,30]. In present study, the MYB binding site involved in flavonoid biosynthetic genes regulation was distinguished in the *DlTIFY7* promoter, suggesting its putative regulation of flavonoid biosynthesis. This requires further investigation.

It is widely accepted that the *TIFY* family genes are involved in diverse biological processes. For instance, overexpression of *OsTIFY11a* led to significantly increased tolerance to salt and dehydration stresses in rice [7]. Similarly, overexpression of *OsJAZ9* could enhance salt and drought stress tolerance [31]. Moreover, overexpression of *GsJAZ2* has been shown to enhance salinity tolerance in *Glycine max* [32]. In rice, OsJAZ6/OsJAZ1 interacts with OsMYC2 and functions in the JA signaling pathway. OsMYC2 directly binds to the promoter of *OsMADS* to activate their expression, which in turn regulates spikelet development [33,34]. *VvTIFY* genes in grape can be induced by cold-, drought-, or salinity stress, as well as JA and ABA treatments [16]. Knockout of *AtTIFY10a* and *AtTIFY10b* decreased alkaline tolerance of mutant *Arabidopsis* during seed germination [35]. In maize, *ZmTIFY16* was induced by low temperature and abiotic stress. Moreover, it promoted root growth by interacting with *ZmMYC2* which is an essential regulator of the jasmonic acid signaling pathway. The overexpression of *ZmTIFY16* could improve salt tolerance of transgenic maize lines [36]. In addition, recent research has revealed that the TIFY family genes also play crucial roles in secondary metabolism. Flavonoids participate in diverse biological functions in plants, including UV photoprotection, defense against pathogen infection, nodulation, auxin transport, pollen fertility, and coloration of flowers and fruits [37,38,39,40]. For example, in *Arabidopsis thaliana*, *TIFY* genes have been shown to regulate the biosynthesis of anthocyanin [2]. In grape, *VvTIFY5A* negatively regulates the synthesis of anthocyanin by repressing the expression of the dihydroflavonol-4-reductase gene, which is essential for anthocyanins synthesis. The overexpression of *VvTIFY5A* reduced the anthocyanin synthesis in grape calli [41]. The apple MdJAZ2 protein functions in the negative regulation of anthocyanin accumulation and peel coloration [10]. The expression of *PpJAZ* genes was negatively associated with anthocyanin content in the pear ‘Mantianhong’ during fruit development [8]. In this study, *DlTIFY* genes were involved in jasmonic acid signaling and response, according to the results of GO and KEGG enrichment analysis. Also, there were several hormone-related cis-acting elements. Consistent with previous studies, ectopic overexpression of *DlTIFY7* in tobacco suppressed the pigment of the corolla. Interestingly, a slightly green color was clearly visible in the corolla and a green ring around the edge of the corolla. This is consistent with a previous study, in which heterologous expression of *ZmTIFY16* in *Arabidopsis* increased chlorophyll contents under drought stress [36]. It has been demonstrated that the flowers’ color is determined by the chlorophyll break down and the synthesis of anthocyanins [42]. This might indicate that *DlTIFY7* plays a role in both anthocyanin and chlorophyll synthesis. Further, it is indicated that the DlTIFY7 protein has relational interaction with MYC2, which is a target of JAZ protein to mediate JA-regulated plant defense and development processes [43,44,45]. These findings collectively suggest that the *TIFY* family genes have extensive and significant functions in various plant biological processes. However, further research is required to uncover the underlying mechanism of DlTIFY for plant pigmentation and explore their potential applications in longan improvement to enhance stress tolerance and metabolite production.

## 5. Conclusions

In previous studies, the function of the TIFY gene family negatively regulating anthocyanin synthesis in many species has been confirmed [2,8,10,41], but the related function of the *TIFY* gene family in Longan has not been reported. In this study, 19 members of the *DlTIFY* gene family were identified in the Longan genome by BLAST and HMMER search and bioinformatic analysis was performed. The prediction results of protein interaction function of longan showed that DlTIFY7 had multiple interacting proteins and might interact with MYC, MYB, and WRKY. Subcellular localization results showed that DlTIFY7 was localized in the nucleus. Overexpression of the *DlTIFY7* gene in tobacco can make the flower crown of tobacco slightly green. This means that *DlTIFY7* may play a negative regulatory role in the anthocyanin synthesis pathway of longan and reduce the anthocyanin content in plants, but the relevant regulatory mechanism needs further study. The aim of this study was to provide a theoretical basis for the synthesis of anthocyanins in Longan and to provide a new target gene for breeding and improvement of Longan varieties.

## Figures and Tables

**Figure 1 biology-14-00364-f001:**
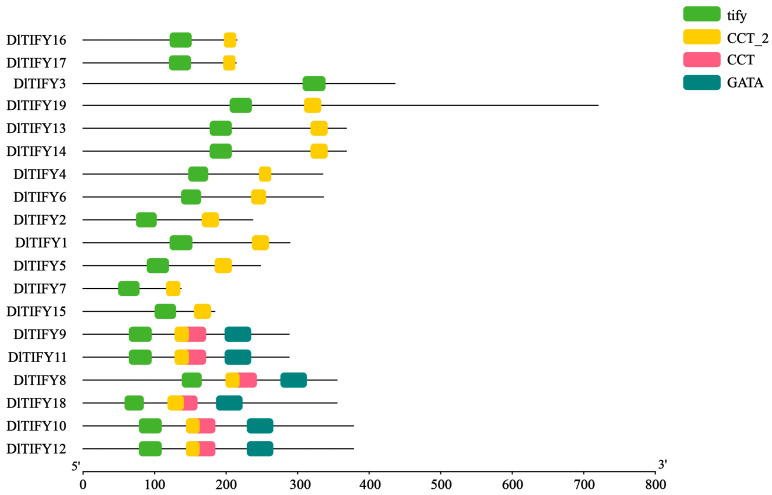
Conserved domains of longan DlTIFY proteins.

**Figure 2 biology-14-00364-f002:**
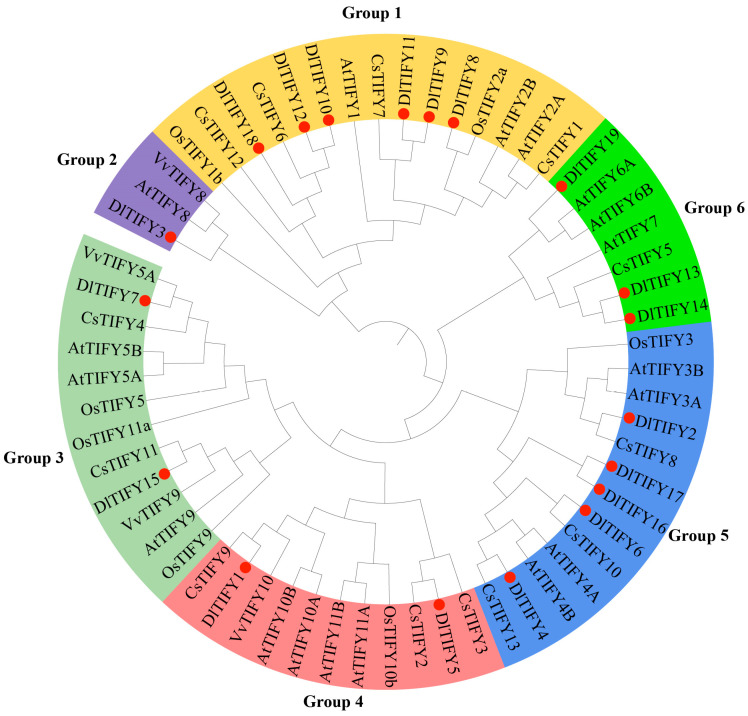
Phylogenetic tree of DlTIFY proteins for longan, grape, orange, rice, and *Arabidopsis*. The tree was divided into 6 groups. Those marked in red circle are members of the Longan TIFY gene family.

**Figure 3 biology-14-00364-f003:**
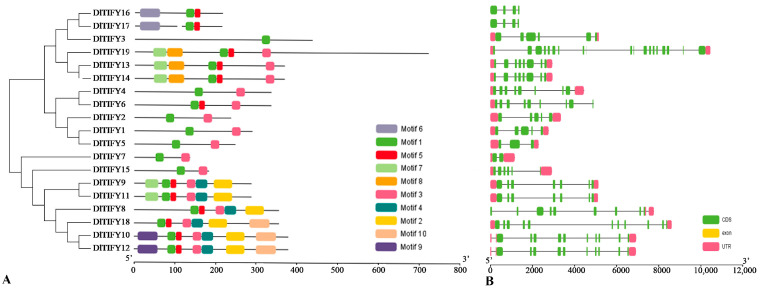
Sequence analysis of DlTIFYs in longan. (**A**) Phylogenetic analysis of longan TIFY proteins and conserved motifs analysis. The motif 1 of TIFY is distributed in each DlTIFY protein. (**B**) Exon and intron structure of longan TIFY genes. UTR, exon, and intron are represented by a red box, green box, and black line, respectively.

**Figure 4 biology-14-00364-f004:**
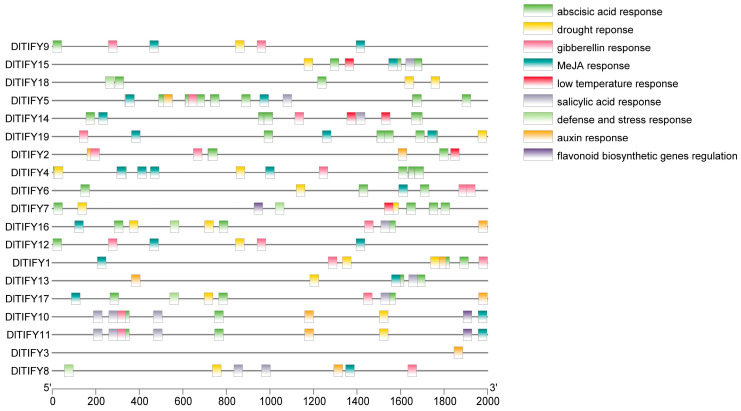
The cis-acting elements of DlTIFYs.

**Figure 5 biology-14-00364-f005:**
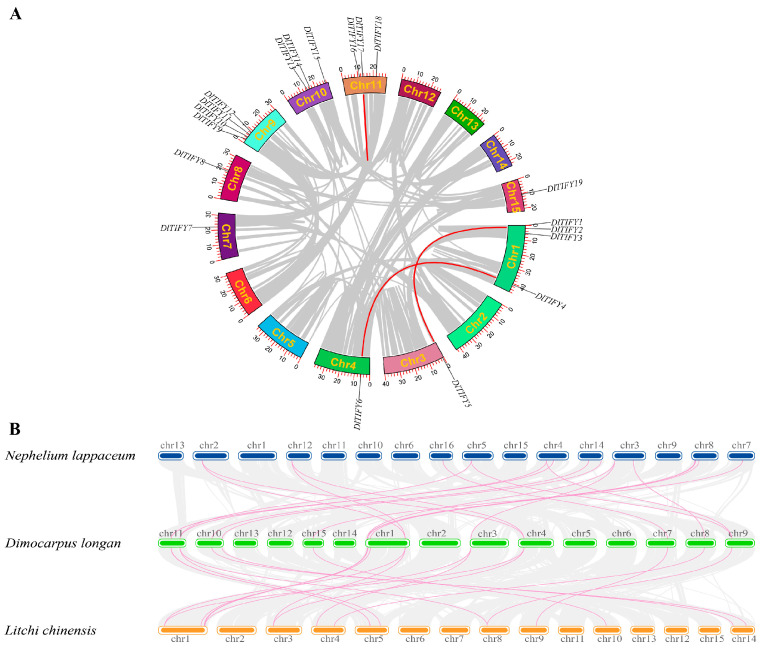
Chromosome distribution and gene duplication. (**A**) Chromosome distribution of *DlTIFY* genes. (**B**) Collinearity analysis of DlTIFYs. Synteny analysis of the *TIFY* genes between longan, litchi, and rambutan of Sapindaceae fruit tree.

**Figure 6 biology-14-00364-f006:**
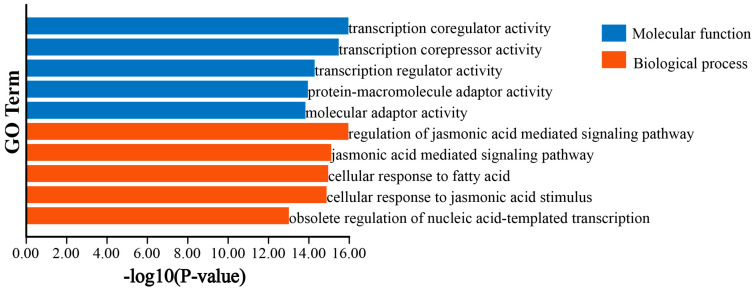
GO analysis of DlTIFYs.

**Figure 7 biology-14-00364-f007:**
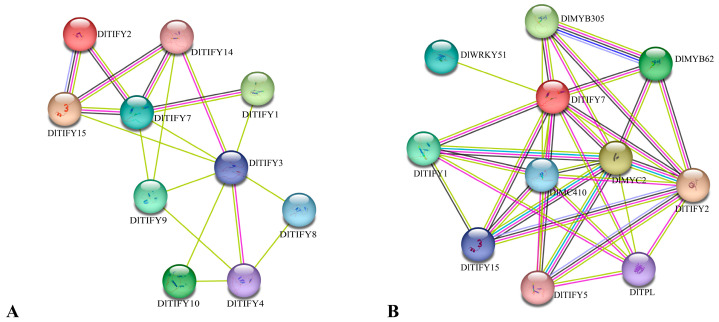
Protein–protein interaction network using STRING database. (**A**) DlTIFY proteins interaction network. Disconnected nodes are hidden. (**B**) The predicted functional partners of DlTIFY7 protein based on their orthologs in *Arabidopsis*, including MYC2, MYB, and WRKY transcription factors.

**Figure 8 biology-14-00364-f008:**
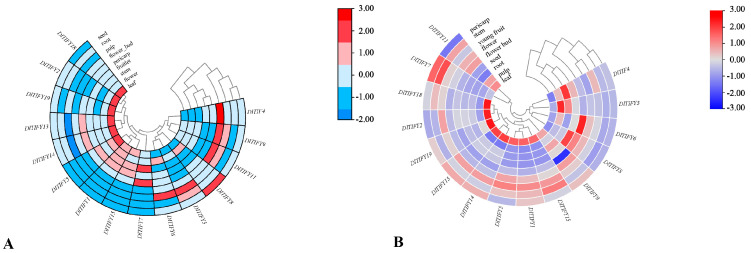
Expression patterns of DlTIFYs in various tissues of ‘Shixia’ (**A**) and ‘Sijimi’ (**B**) longan. The heatmap was generated using log2-transformed values. The color-scale indicates red as higher expression levels and blue as lower expression levels.

**Figure 9 biology-14-00364-f009:**
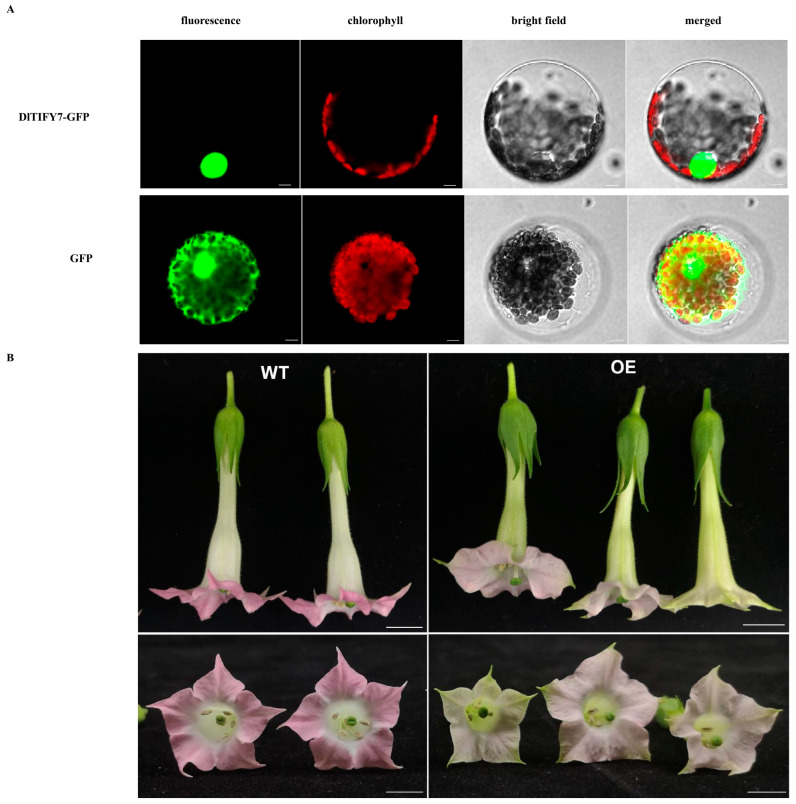
Subcellular localization of the DlTIFY7 protein (**A**) and phenotypes of flowers from the control and *DlTIFY7* overexpressing tobacco plants. (**B**) The DlTIFY7-GFP and empty plasmids (GFP) were observed with UV light, red light, and white light. The 35S::DlTIFY7-GFP fusion protein was localized in the cell nucleus. Bars of 20 μm. The corolla of *DlTIFY7* over-expression (OE) tobacco showed a more pronounced green color than that of wild-type (WT). Bars of 1 cm.

**Table 1 biology-14-00364-t001:** The TIFY genes in *D. longan*.

Gene Name	Gene ID	Chromosome Location	CDS/bp	Size/aa	pI	Molecular Mass/ku	Subcellular Localization
*DlTIFY1*	Dil.01g002090	Chr1: 1595544~1598311	867	288	8.94	30,911.09	Nucleus
*DlTIFY2*	Dil.01g002730	Chr1: 2125173~2128525	711	236	5.18	24,954.09	Nucleus
*DlTIFY3*	Dil.01g005880	Chr1: 4817410~4822560	1308	435	8.98	46,050.99	Nucleus
*DlTIFY4*	Dil.01g031450	Chr1: 40844418~40848860	1005	334	8.68	35,903.26	Nucleus
*DlTIFY5*	Dil.03g000830	Chr3: 675367~677656	744	247	9.08	26,945.32	Nucleus
*DlTIFY6*	Dil.04g007360	Chr4: 5796350~5801241	1008	335	8.64	36,650.41	Nucleus
*DlTIFY7*	Dil.07g012790	Chr7: 22023102~22024262	411	136	9.60	15,454.49	Nucleus
*DlTIFY8*	Dil.08g009850	Chr8: 19156311~19164074	1065	354	8.41	38,771.16	Nucleus
*DlTIFY9*	Dil.09g007890	Chr9: 6380164~6385297	864	287	6.15	31,263.42	Nucleus
*DlTIFY10*	Dil.09g007900	Chr9: 6387450~6394375	1134	377	4.69	41,380.79	Nucleus
*DlTIFY11*	Dil.09g008150	Chr9: 6682786~6687897	864	287	6.15	31,263.42	Nucleus
*DlTIFY12*	Dil.09g008160	Chr9: 6690088~6697001	1134	377	4.69	41,380.79	Nucleus
*DlTIFY13*	Dil.10g008370	Chr10: 16658489~16661430	1104	367	8.86	38,651.59	Nucleus
*DlTIFY14*	Dil.10g008390	Chr10: 16722598~16725548	1104	367	8.86	38,574.55	Nucleus
*DlTIFY15*	Dil.10g020310	Chr10: 27282532~27285457	552	183	9.30	20,614.34	Nucleus
*DlTIFY16*	Dil.11g007770	Chr11: 12937525~12938905	645	214	9.15	24,279.82	Nucleus
*DlTIFY17*	Dil.11g007950	Chr11: 13210964~13212321	642	213	9.30	24,185.7	Nucleus
*DlTIFY18*	Dil.11g015490	Chr11: 22161800~22170409	1065	354	5.03	39,204.58	Nucleus
*DlTIFY19*	Dil.15g013250	Chr15: 10560028~10570474	2160	719	7.21	77,478.53	Cytoplasm, Nucleus

## Data Availability

The raw data supporting the conclusions of this article will be made available by the authors on request.

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
