# Peer review of "Genome-Wide Identification of the TIFY Family in Longan and Their Potential Functional Analysis in Anthocyanin Synthesis"

_biology, 2025, doi:10.3390/biology14040364_

Round 1

Reviewer 1 Report

Comments and Suggestions for Authors

The study identifies 19 DITIFY genes in the Longan genome and thoroughly characterizes them in terms of chromosomal distribution, phylogenetic relationships, conserved domains, gene structure, and cis - acting elements. An important aspect of this study is the investigation of DITIFY7’s potential role in anthocyanin synthesis. The observation of altered flower color in transgenic tobacco serves as direct functional evidence, making this a promising area for further research. However, although the study is significant, the following issues need to be addressed prior to publication:

  1. In the current manuscript, the reason for choosing anthocyanin synthesis as the main functional focus for Longan TIFY genes is not clearly explained. Adding one or two sentences in the introduction about anthocyanin - related characteristics of Longan, such as pericarp pigmentation, would strengthen the rationale.
  2. It is interesting to note that DITIFY19 is predicted to localize to both the cytoplasm and nucleus, while all other DITIFYs are predicted to be nuclear. However, this has not been explained. Is this due to a limitation of the prediction tool, or does it reflect a biological difference, such as dual functionality? This inconsistency undermines the confidence in the localization predictions. Please discuss this discrepancy in the results or discussion section.
  3. The reference format is inconsistent throughout the paper.
  4. It is advisable to highlight the genes of the target species in the phylogenetic tree.
  5. The DlTIFY family is classified into three subclades based on conserved domains. It is recommended to illustrate this in the domain analysis figure.
  6. There are errors in the use of spaces. There are multiple spaces between some words, and spaces are missing around punctuation marks.
  7. The grammar of the paper requires improvement.

Reviewer 2 Report

Comments and Suggestions for Authors

This manuscript presents a well-executed study on the DlTIFY gene family in longan. Through comprehensive bioinformatics analyses, the authors identified 19 DlTIFY genes from the longan genome data. They also found that the expression of DlTIFY7 in the stem and pericarp was significantly higher than that in other tissues. These findings suggest that DlTIFY7 may serve as a novel candidate transcription factor negatively regulating anthocyanin synthesis in longan. I believe this study can provide valuable insights for future research in this area. However, major revisions are necessary due to the following shortcomings.

Main shortcomings:

  1. Line 45: In the sentence of “The first TIFY family genes was identified in,” “family genes” is plural, so it should be changed to “The first TIFY family genes were identified in”
  2. Lines 54-55: The sentence should be modified to “Whereas, AtTIFY4a and AtTIFY4b wereinvolved in the process of leaf development [4].”
  3. Line 57: The sentence “The TIFY proteins were also play a role in abiotic stress and phytohormone responses.” contains a grammatical error. It should be revised to “The TIFY proteins also play a role in abiotic stress and phytohormone response.”
  4. Line 82:The first letter of the word “dentification” should be capitalized.
  5. In the section of Materials and Methods, version information for the analytical tools used in the study should be included. The authors should carefully check and provide version details for tools such as TBtools, ClustalW, MCScanX, and PlantCARE.
  6. Line 87. The website for Pfam is incorrect. The pfam.sanger.ac.uk website is no longer supported at The Wellcome Sanger Institute. Pfam data and new releases are available through InterPro (https://www.ebi.ac.uk/interpro/).
  7. Lines 114-115. “were used” should be changed to “was used”
  8. Lines 130-131: The correct URL for the STRING (Search Tool for the Retrieval of Interacting Genes/Proteins) database is: https://string-db.org/
  9. The image of Figure 1 is unclear with low resolution. It would be better to enlarge it a bit, and separate sections A and B.
  10. Line 212: “two pair of genes” should be “two pairs of genes”
  11. Lines 213-214: In the sentence “Three segmentally duplicated pairs was found by collinearity analysis of the DlTIFY genes,” the word “was” should be revised to “were”
  12. In the section of Discussion, the analysis of tissue-specific expression patterns of DlTIFY genes (lines 366-368) only describes the high expression in leaves and pulp. The authors should explore the biological significance of this expression pattern. Specifically, why is DlTIFY3 highly expressed in the pulp? At what stage of fruit development might it play a role?
Comments on the Quality of English Language

The manuscript contains numerous grammatical errors.  Relevant comments have been provided in the "Comments and Suggestions for Authors
".

Reviewer 3 Report

Comments and Suggestions for Authors

In this paper, the authors identified 19 DlTIFY genes from the longan genome and performed bioinformatic analyses. Among these genes, DlFITY7 was predicted to be regulated by JA and to contribute to flavonoid synthesis via MYC, MYB and WRKY transcription factors. Furthermore, analysis of DlFITY7-overexpressing tobacco plants showed that this gene may negatively regulate anthocyanin synthesis. The analysis has been carried out using well-established methods and provides useful information. However, there are many points of improvement that could be made to improve the description of the paper. My detailed comments are as follows.

Major points

Throughout the textSome abbreviations have no explanation or have an explanation outside the first part of the text (e.g. L. 22-23, L. 40, L. 170, L. 200, L. 299, etc.) and should be corrected. All figures are too low resolution to be visible. This should also be corrected.

L. 123-137: No one can verify reproducibility with these descriptions. Be sure to describe the necessary information, such as detailed methods for protoplast extraction, filters and wavelengths used for fluorescence observation, genome or RNA extraction and cDNA synthesis, DNA polymerases used for PCR, restriction enzymes used for vector construction, antibiotics used for selection of transformants, etc.

L. 287: In this section, DlTIFY7 is chosen but there is no explanation of why. Authors must explain why this gene was chosen from 19 DlTIFY genes.

L. 313-326: This section is often poorly written and contains repetitions of sentences similar to those in the introduction. In my opinion, this part should be deleted.

L. 327-345: This section contains confusing wording and over-speculation. It is interesting to note that the longans do not have PPD gene despite being dicotyledonous, but the reasons for this are not convincing based on what is described here. Speculations should be kept to a minimum as they will only confuse the reader. In addition, the author considers the relationship between the number of TIFY genes and genome size, but information on the genome size of longan should be described.

L. 363-365: Missing mention of biological roles of TIFY genes in Arabidopsis thaliana and Oryza sativa. Especially, DlTIFY7 is highly expressed in stem, so a comparison with the function of AtTIFY5a is essential.

L. 372-375: I could not find any mention of these results in the Results section. Since the function of these transcription factors differs depending on the isoform, precise descriptions are needed.

L. 386-399: Although this part describes the relationship between TIFY and stress tolerance, no experiments on stress tolerance were conducted in this paper. What is the intention of the detailed description of stress tolerance?

L. 417-418: Please provide appropriate references on the relationship between chlorophyll degradation and anthocyanin synthesis.

L. 430-437: Conclusions section simply summarizes the results of this paper. It should include future perspectives, for example, comparisons with TIFY genes in other plants and what the information obtained in this paper could be useful for.

Minor points

Table 1: It is recommended to not abbreviate “Nuc”, but to write it as “Nucleus”.

Figure 2: It would be better to arrange the motif in numerical order.

L. 208: ~~in Chromosome (chr) 1 and chr 9. The chr 10 ~~

Figure 8: No explanation of what “WT” and “OE” are in legend.

References: To be unified according to the format specified by MDPI.

Comments on the Quality of English Language

Please carefully check the entire text for any errors such as italics or forgotten spaces, etc. 

Round 2

Reviewer 3 Report

Comments and Suggestions for Authors

The authors have responded politely to my comments, but a few points remain uncorrected. In addition, spelling errors (L. 135, etc.) and missing spaces (spaces before brackets, etc.)between words are still found, and should be corrected in the final version.

L. 381 – 386: The roles of OsTIFY genes are missing. The authors argue that the expression patterns of the TIFY genes are related to their biological roles. If authors want to mention that there is a link between expression patterns and physiological functions in different plant species, the example of rice should be added.

L. 403 – 405: I pointed out earlier that the descriptions for DlMYC2, DlMYB305 (and DlMYB62), DlWRKY51 should be described in the results section. In the results section, be sure to state the number of the transcription factors. Then, to correct the errors in English in this part.

Comments on the Quality of English Language

Spelling errors (L. 135, etc.) and missing spaces (spaces before brackets, etc.)between words are still found, and should be corrected in the final version.
